# Towards Optical Imaging for Spine Tracking without Markers in Navigated Spine Surgery

**DOI:** 10.3390/s20133641

**Published:** 2020-06-29

**Authors:** Francesca Manni, Adrian Elmi-Terander, Gustav Burström, Oscar Persson, Erik Edström, Ronald Holthuizen, Caifeng Shan, Svitlana Zinger, Fons van der Sommen, Peter H. N. de With

**Affiliations:** 1Department of Electrical Engineering, Eindhoven University of Technology, 5600 MB Eindhoven, The Netherlands; s.zinger@tue.nl (S.Z.); fvdsommen@tue.nl (F.v.d.S.); P.H.N.de.With@tue.nl (P.H.N.d.W.); 2Department of Clinical Neuroscience, Karolinska Institutet, Stockholm SE-171 46, Sweden & Department of Neurosurgery, Karolinska University Hospital, SE-171 46 Stockholm, Sweden; adrian.elmi-terander@sll.se (A.E.-T.); gustav.burstrom@ki.se (G.B.); oscar.persson.1@ki.se (O.P.); erik.edstrom@sll.se (E.E.); 3Philips Healthcare, 5684 PC Best, The Netherlands; ronald.holthuizen@philips.com; 4Philips Research, High Tech Campus 36, 5656 AE Eindhoven, The Netherlands; Caifeng.Shan@gmail.com

**Keywords:** optical sensing, spinal surgery, image processing, image analysis for markerless tracking, patient tracking, image-guided surgery

## Abstract

Surgical navigation systems are increasingly used for complex spine procedures to avoid neurovascular injuries and minimize the risk for reoperations. Accurate patient tracking is one of the prerequisites for optimal motion compensation and navigation. Most current optical tracking systems use dynamic reference frames (DRFs) attached to the spine, for patient movement tracking. However, the spine itself is subject to intrinsic movements which can impact the accuracy of the navigation system. In this study, we aimed to detect the actual patient spine features in different image views captured by optical cameras, in an augmented reality surgical navigation (ARSN) system. Using optical images from open spinal surgery cases, acquired by two gray-scale cameras, spinal landmarks were identified and matched in different camera views. A computer vision framework was created for preprocessing of the spine images, detecting and matching local invariant image regions. We compared four feature detection algorithms, Speeded Up Robust Feature (SURF), Maximal Stable Extremal Region (MSER), Features from Accelerated Segment Test (FAST), and Oriented FAST and Rotated BRIEF (ORB) to elucidate the best approach. The framework was validated in 23 patients and the 3D triangulation error of the matched features was <0.5 mm. Thus, the findings indicate that spine feature detection can be used for accurate tracking in navigated surgery.

## 1. Introduction

Surgical navigation systems provide a reliable image-guided solution for complex interventions such as spinal surgery [1,2,3]. An important step in spinal fixation surgery is the placement of pedicle screws. Safe placement of these screws, requires high accuracy as the surgical risks include damage to vital neurological and vascular structures in close anatomical relation to the pedicles [4]. The traditional free-hand technique relies on a combination of anatomical landmarks, pre-operative imaging and use of X-ray fluoroscopy [5,6,7]. However, the accuracy with this technique is greatly dependent on the surgeon’s expertise. In a meta-analysis, Gelalis et al. report that the percentage of screws with cortical violations > 4 mm was 1.0–6.5% [8]. The addition of surgical navigation systems to spinal fixation surgery aims at improving these numbers for a more accurate and safe surgery [9,10,11,12,13,14,15]. Current commercially available spinal navigation systems utilize an indirect method to track the spine, and work on similar principles, which are illustrated in Figure 1 [16,17,18].

The indirect method uses optical hardware to identify a dynamic reference frame (DRF) attached to the patient’s spine [19,20]. DRFs typically consist of reflecting spheres on a metal star with a known positional geometry, which is recognized and used by the navigation system for patient tracking [21]. To initiate tracking, the navigation software is provided with information on the patient position in relation to the markers through a user feedback procedure or intraoperative imaging [18,20,22]. Hence, the navigation is accurate only as long as the spatial relationship between the markers and the patient remains undisturbed. Imaging data is integrated through intraoperative 3D scanning or co-registration to preoperative 3D scans [19,23]. Since a DRF is attached to a single vertebra, it may produce navigational errors if there is movement within the spinal column during surgery [3,19]. To address this problem, we have previously reported on an augmented reality surgical navigation (ARSN) system using non-invasive optical markers, attached to the skin and detected by live video cameras [10,14,15,20,24,25].

In contrast to reference frame-based solutions with a single attachment point, the optical skin markers are more uniformly distributed and can potentially provide accurate navigation over a larger part of the spine. Nonetheless, any type of marker attachment can be dislodged or obscured, resulting in loss of navigational feedback. Markerless tracking solutions have been used experimentally on phantoms in other surgical fields, however validation in complex spine surgery in clinical cases is lacking [26,27]. A device using optical surface matching, although not primarily designed for medical use, is the Microsoft Hololens. Experimental use in non-medical models have demonstrated an accuracy ranging from 9–45 mm depending on distance to the object [28]. In a spine phantom study an accuracy of roughly 5 mm was achieved [29]. These accuracies are not good enough for clinical use. In this study, we present a technology to omit the usage of markers altogether while maintaining surgical accuracy. Using the ARSN system previously presented, we developed a technology utilizing the integrated live video cameras to directly track the patient’s spine feature (Figure 2). The technology relies on the system to consistently recognize key features of the patient’s spine. Once two descriptive feature points are detected it is possible to estimate the parameters of similarity transformation [30], so that the spine movement can be detected and corrected for surgical guidance. In order to find salient points along the spine, anatomical spine features detection is needed. The key part is to find correspondences between at least two camera views, by employing feature detection algorithms and using them to perform stereo matching and 3D error assessment. The feasibility of spine feature detection, using hand-crafted local feature detection algorithms, was previously investigated, and the feature tracking algorithm was successfully applied [31].

The approach aims to detect local invariant image regions (in the form of blobs or corners), to compute descriptors evaluated in the context of matching of a scene under different views [32]. To extend our work, several rotation and affine-invariant local feature detection algorithms were evaluated, to assess the feasibility of the methodology. Benchmarking was performed against four popular feature extraction techniques: the Maximal Stable Extremal Region (MSER) proposed by Matas et al. [33], Speeded Up Robust Feature (SURF) [34], Features from Accelerated Segment Test (FAST) proposed by Rosten and Drummond in [35], and Oriented FAST and Rotated BRIEF (ORB), which all offer good computation cost and matching performances. The above-mentioned techniques were employed for spine features detection, resulting in different regions of interest used for feature matching. Corresponding features were triangulated to assess the 3D feature location and estimate the triangulation error which can potentially facilitate spinal column transformation estimation and compensation. Aimed at improving the workflow and accuracy of surgical navigation systems, this study illustrates the feasibility of direct spine detection and matching in multi-view optical imaging. The advantage of the current framework is that an extensive comparison has been performed between four different local feature detection algorithms, aiming at higher 3D matching accuracy. To the best of our knowledge, this is the first work to apply 3D anatomical feature detection for image-guided spine surgery. The contributions are: (1) a computer vision framework for preprocessing of optical spine images, detecting and matching local invariant image regions for two different image views, (2) a comparison of four feature detection algorithms, SURF, MSER, FAST, and ORB to elucidate the best approach for reliability and accuracy, (3) validating the framework on surgical images acquired in 23 patients to assess the 3D triangulation error of the matched features.

## 2. Materials and Methods

This study was designed to build a robust framework for spine feature detection and matching by using optical imaging as illustrated in Figure 3. Spine feature detection aims to find correspondences between pairs of multi-view images over different time frames. Feature descriptors are extracted and matched based on their similarity. Image projections, from two camera views, were used to determine the 3D location of detected features in the human spine, by means of 3D triangulation. Based on the found matches, 3D spine feature locations and incorrect correspondences were discarded. Furthermore, feature correspondences were visually identified to ensure a better validation.

The image acquisitions were primarily made for an optical, marker-based, navigation system (ARSN). The navigation system was used in a series of twenty-three patients undergoing augmented reality navigated spine surgery. The study was approved by the local ethical board. All patients signed informed consent. Per each unit of time, images from two camera views, where the spine was fully visible, were analyzed (Figure 2). The images from the other two cameras were not considered, since the spine was not visible.

The proposed approach, described in detail in the following subsections, entails an application of a standard computer vision framework, which aims to design and match spine features between two camera views.

### 2.1. Image Preprocessing for Improved Feature Matching

This section presents the preprocessing phase required for enhancing the spinal cord and the feature matching from different image views.

The first step for an exhaustive image matching was image rectification, which resamples pairs of stereo images taken from different viewpoints, in order to produce matched epipolar projections. After applying a pair of 2D projective transformations to the two image views, the epipolar lines matched between views and there was no *y* disparity [36]. These are projections in which the epipolar lines were oriented in parallel with the *x*-axis and match up between views, and as a result disparities between the images were in the *x*-direction only [36] (see Figure 4). Notably, the horizontal *x*-axis and the vertical *y*-axis were identified depending on the respective 3D positions of the cameras, where the *x*-direction was the connecting line between the two cameras.

The applied image rectification algorithm consisted of: (1) identifying a set of matched points between two images, (2) computing the fundamental matrix *F* which expressed the correspondence between the points in the stereo image pair, (3) selecting a projective transformation *H* that mapped the epipole *e* and $e^{\prime }$ to the point at infinity, (4) finding the matching projective transformation $H^{\prime }$, (5) resampling the first image according to the projective transformation *H* and the second image according to the projective transformation $H^{\prime }$. The matching projective transformation *H* that minimized the disparity along the horizontal axis was defined by: (1)$$\sum_{i}d{\left(Hx_{i},H^{\prime }x_{i}^{\prime }\right)}^{2}=0.$$
where *d* represents the sum-of-squared distances. After image rectification, the search for matching points between different image views, was significantly simplified by the simple epipolar structure and by the overall correspondence of the two images [36]. Then, basic preprocessing methods were applied to enhance the desired features by adding a contrast limitation. In this approach, contrast limited adaptive histogram equalization (CLAHE) [37] was chosen, since the difference between the contrast distinctions in different areas of the rectified images was significant. A contrast limit of 0.03 and a uniform histogram shape were chosen. Then, an image thresholding was performed, to create a mask which was multiplied to each contrast enhanced image and applied to segment the spine, as shown in Figure 5. The image thresholding was performed by using the Otsu method. Lastly, the area around the spine was manually cropped for further analysis. The manual cropping was only needed for limiting the feature detection to the spinal area. Afterwards, the detected features were projected back to the two image views preserving their original 2D locations in the image plane. This manual extraction constituted a limitation for real-time use of the system. However, this study was only an initial investigation on the feasibility of using spine features for 3D spine position detection during navigated surgery and did not aim to solve the issue of real-time application.

### 2.2. Spine Feature Detection

Robust feature detection and local image descriptors are the prerequisites for finding correspondences between two or more images. In this application, they allow the assessing of the 3D location of the spine and eventually, motion correction. We have previously adapted SURF and MSER to study the feasibility of anatomical landmark detection on skin or spinal column [31,38].

In this work, four feature detection algorithms, SURF, MSER, FAST and ORB were evaluated to assess the feasibility of detecting salient features in the spine without using skin markers.

The basic framework for a feature detection algorithm is based on the principles of scale-space representation, key-point localization, orientation assignment, and key-point descriptor extraction. The SURF algorithm has the advantage of returning reproducible features under different viewing conditions. The method employs the approximation of the Hessian matrix determinant for each pixel in the image for the detection of interest points. The Hessian matrix is based on second-order derivatives of the image signal at the position of *x* in scale σ, which is specified by:(2)$$H\left(x,\sigma \right)=\left[\begin{array}{cc} L_{xx} & L_{xy}\\ L_{yx} & L_{yy} \end{array}\right].$$
where, $L_{xx}$, $L_{xy}$ and $L_{yy}$ are the second-order derivatives of the Gaussians of the image in *x*- and *y*-directions at point *x*. After the Hessian matrix calculation, the transformed image is acquired by computing the approximation of the determinant of the Hessian matrix which is specified by:(3)$$det\left(H_{approx}\right)=L_{xx}L_{yy}-{\left(0.9\cdot L_{xy}\right)}^{2}.$$

The constant 0.9 is part of the approximation. Then, the image is supplied to a pyramid of filters at different scales. Each pixel is compared to its neighbors and it will be returned as a feature point only if it is the maximum or minimum of all these surrounding points. Haar wavelets are used for detecting the dominant direction. To this end, the Haar wavelet responses are computed in the horizontal and vertical directions, for all feature points. These responses are forming a new vector. The direction of the longest vector is selected as the dominant direction of the feature point. The dominant direction of each of the interest points is found in order to support rotation-invariant matching. Finally, the SURF descriptor is returned as a 64-dimensional vector, obtained by summing the Haar wavelet coefficients over a 4×4 pixel area around the keypoint, which will be used for feature matching. An image is analyzed at several scales, to extract interest points from both global and local image details.

Alternatively, MSER is a blob detection algorithm which captures salient features that are invariant to rotation, scaling and affine transformation, for thresholding the image at various intensity values between 0 and 255. While the threshold is increasing, a few coherent areas will gradually appear. The maximum stable extreme region is determined by the threshold that gives the smallest change (and the maximal stability) in the growing area. The main steps of the algorithm can be summarized as follows: (1) thresholding the image by scanning over an intensity range from black to white, (2) extracting the connected regions (extreme regions) and approximating with the bounding ellipse, (3) finding the threshold corresponding to the maximally stable extreme region. The identified regions represent the feature points. It should be noted, that extreme regions have the property of being affine invariant [33]. Thus the algorithm is not affected by image warping and skewing and it performs well with view-point change. MSER, as a multi-scale detection approach, achieves good performances for both small and large homogeneous structures.

Different illumination conditions affect spine visibility for the video cameras. Poor lighting conditions can decrease the number of matched features and mislead the spine movement estimation. The analysis was extended to enlarge the benchmarking overview and improve the number of detected and matched features. Therefore, FAST and ORB were adopted as additional feature detection algorithms since both show good feature detection performances. FAST is a corner detection method which compares the brightness with the intensity level of pixels included in a threshold. The method compares pixels on a circle of fixed radius around a point *p* (candidate interest point). A point is classified as a corner only if a large set of contiguous pixels (i.e., 16 pixels) on a circle fixed radius are all brighter or darker than the candidate point plus a threshold T [35]. If at least three pixel values are not above or below the intensity level of the candidate point, *p* is not an interest point. If there are at least three pixels above the intensity level of *p* plus a threshold T, the algorithm checks if all 16 contiguous pixels fall in the same criterion. The main goal of the FAST algorithm is to develop an interest point detection method for real-time applications [35]. The method achieves the goal of having a detector, several times faster than other existing corner detectors. However, it has the main drawback of detecting multiple features adjacent to one another [35].

ORB is a combination of a FAST keypoint detector and BRIEF descriptor [39,40]. It uses FAST as feature detector and BRIEF to extract the descriptor. Since FAST does not include an orientation operator, ORB computes the moment of the patch surrounding the feature points. BRIEF is then used to extract the descriptor around the feature point, by performing binary tests between pixels in a smoothed image patch [39]. Since BRIEF performs poorly with rotation, ORB rotates the descriptor according to the keypoint orientation, adding the rotation invariance to the descriptor. The ORB feature is a binary-based feature with several advantages, it is computationally more efficient and easier to store than the vector-based features [40].

Table 1 shows the parameter values used for each feature detector algorithm to detect spinal features. These parameter values are chosen to design and detect features on the spine that are sufficiently reliable to ensure feature matching along the vertebral column.

### 2.3. Spine Feature Matching of Multi-View Images

The matching algorithm finds correspondences between the descriptors which have been extracted for two image views. For this, the salient feature points detected by the SURF, MSER, FAST, and ORB algorithms, were used to compute feature descriptors and the descriptors were matched to find the corresponding features between the two views.

Normalized cross-correlation was chosen as feature matching metric and two vectors containing indexes of the matched features were provided as final output. After matching, outliers were discarded if they did not satisfy the epipolar constraint. The outliers are mismatched feature points and appear because of the similarity between the descriptors. As explained in the preprocessing step, after image rectification, the epipolar lines for corresponding points are collinear. Hence, the epipolar constraint was used to remove invalid feature points where an incorrect correspondence was found [31].

### 2.4. 3d Stereo Triangulation

Three-dimensional triangulation allows us to determine the 3D position of a point *X* by using the 2D projection of *X* in two image views, $x_{1}$ and $x_{2}$ (see Figure 6). Given the two image views $I_{1}$ and $I_{2}$ and a fundamental matrix *F*, the epipolar line in $I_{2}$ that corresponds to a point in $I_{1}$ and vice versa can be computed. The epipolar line is the intersection between the plane defined by the two camera centers and the image plane of $I_{2}$. For a more detailed description of this procedure we refer the reader to an extensive work by Hartley and Zisserman on this topic [36].

Figure 6 shows an ideal triangulation, where $C_{1}$ and $C_{2}$ define the 3D positions of Camera 1 and 2, respectively. Briefly, the intersection of lines $C_{1}x_{1}$ and $C_{2}x_{2}$ gives the 3D position of *X*. However, the presence of noise (e.g., lens-distortion noise) can lead to inaccurate intersection. In this study, for minimizing the 3D error in a simplest and effective way, the 3D position of *X* was chosen as the midpoint between back projected image points. The triangulation error is calculated as the minimum distance between the lines $C_{1}x_{1}$ and $C_{2}x_{2}$, and captured in the line segment $V_{p}$ (Figure 6). The length of the line segment $V_{p}$ is the triangulation error. The mean of the triangulation errors of the matched features is then used to assess the method.

## 3. Results

In this study, 23 open spinal surgery cases were analyzed. The aim of the study was to assess the feasibility of detecting and matching spinal features in multi-view stereo images (Figure 7), and to compare four different feature detection algorithms. Figure 8 visualizes an example of spinal feature detectors for two image views for a single patient, when the above-described algorithms were applied.

The obtained 3D accuracy of each method was measured with the triangulation error for each spinal feature and is depicted in Figure 9. For each unit of time, the 3D triangulation error of the salient feature points matched in different viewpoints from the two cameras, was calculated, as described in Section 2.4. The average triangulation error (±standard deviation) when using SURF, MSER, FAST, and ORB was calculated for each patient. The triangulation errors for SURF, MSER, FAST, and ORB were 0.38±0.5, 0.38±0.6, 0.41±0.07, and 0.43±0.04 mm, respectively.

The lowest mean triangulation error was observed when SURF and MSER were as local feature detection algorithms, slightly outperforming the other evaluated methods. It should be noticed that the results obtained for Patient 6 were based on only one frame, due to the limited recording time with sufficient illumination. However, for this frame a total number of matched inliers equal to 19, 15, 21, 129, was found when SURF, MSER, FAST, and ORB were applied, respectively. The statistics for the matched spinal features are detailed in Table 2.

It is important to consider brightness changes which may lead to a lower number of matched feature points. The performances of different descriptors, and their capability to detect and match a reasonable number of landmarks in the spine, were evaluated. In a quantitative comparison, it was observed that ORB detected the highest number of matched features, as shown in Table 2. Nevertheless, when the illumination conditions decreased, the algorithm achieved a minimum number of inliers equal to unity, which could not guarantee a spine transformation estimation. Instead, a reasonable number of average matched features were detected when SURF and FAST were used (minimum number of inliers equal to 2). A visual comparison of both methods in Figure 10 shows that FAST detected less dense corner features compared with SURF.

The median error and the interquartile ranges, showed the same variability among the patients when using FAST or ORB for 3D triangulation error (Figure 11). The variability and the outliers may have been caused by either brightness differences during the recording, or limited visibility of the spine. For each recording, all frames where the spine was not visible by two cameras simultaneously were discarded. Four patients reported a low number of analyzed frames (Patient 4, Patient 6, Patient 12, Patient 13). The total number of analyzed image pairs was 324 for all the patients, with an average of 17 pairs of frames per patients. For better visualization of the individual evaluated results, the cumulative distribution functions (CDFs) of the triangulation errors were calculated (Figure 12). The dotted area delimits the 95% confidence interval for each CDF. Plotting the cumulative distribution function shows that more than 95% of the analyzed frames had a triangulation error lower than 0.5 mm when SURF, FAST, or ORB was used. For MSER, however, the corresponding number was 84%, likely explained by the larger variability in triangulation errors for MSER (Figure 12).

### Computation Times

The CPU times for the core tasks of the algorithms (preprocessing, feature detection and matching) were reported using an Intel(R) Xeon(R) E5-1650 CPU at 3.60 GHz.

The global average computation times required for feature detection for the studied algorithms were 0.05, 0.22, 0.03, and 0.16 s per frame, when SURF, MSER, FAST and ORB were employed respectively (Table 3). While FAST was slightly faster than SURF, MSER and ORB were the slowest methods (four times slower in average compared to SURF and FAST) (Figure 13). Given the performance achieved by SURF and FAST (20 and 33 fps, respectively) on the tests reported above, these feature detection algorithms are recommended for future applications.

The most time-consuming task in the preprocessing step was the region growing method needed to get the final spine segmentation as a binary image. The average preprocessing time was 0.98 s (Table 3, Figure 13).

In this study, the experiments have been conducted using the existing navigation system (the Augmented Reality Surgical Navigation (ARSN)). To employ the findings in a clinical application and in real-time, will require improvements on the navigation hardware to better visualize the spine anatomy. A dedicated set-up will decrease the computation time and a real-time spine feature localization will be possible.

## 4. Discussion

During spinal surgery, optical imaging could potentially offer an attractive solution for non-invasive patient tracking. Real-time surgical guidance implies the use of either DRFs or markers, but the conventionally used tracking systems for spinal navigation have several limitations. One limitation may be insufficient fixation between the DRF and the spinous process, leading to incorrect navigation information. A second limitation is the inadvertent interruption of the camera views of the DRF during the navigation procedure, leading to loss of patient tracking [41,42]. Third, if the DRF is attached to a certain vertebra, movement between consecutive vertebrae increase the possible error for each level away from this “index vertebra” [3,19]. To address these issues, we have developed an augmented reality surgical navigation system (ARSN), which uses optical adhesive skin markers for accurate patient tracking [43]. The overall mean technical accuracy in that study, was 1.65±1.24 mm. To improve on these results, a study using hyperspectral imaging for skin feature detection was performed and reached an accuracy <0.5 mm [44]. However, while skin feature detection offers a good solution in minimally invasive spine surgery, it is impractical in open surgery cases. Moreover, in the cervical spine movements between adjacent vertebrae may go undetected if only skin surface tracking is used. To overcome these obstacles, this work applies a similar methodology to track the spine itself.

Spine feature tracking offers an extension and an improvement of current tracking systems, aiming for optimal patient motion compensation and reliable surgical guidance. Having a tracking technique that relies on features directly related to each vertebra in the surgical field has the potential to be more accurate than dynamic reference frames, which only provide tracking of a single vertebra, or patient tracking techniques with an indirect relationship to the movement of the spine [3,19].

In this paper, we have presented an application of SURF, MSER, FAST, and ORB algorithms, which are used to design and detect spine features. The resulting triangulation errors, well below 1 mm, are clinically acceptable and proves the feasibility of assessing the 3D vertebra location using direct spine features [45]. The results here, form the basis for direct tracking of the spine and improving the accuracy of the navigation and thereby facilitating accurate surgery. An important criterion for achieving an accurate feature matching is the number of extracted features. Although SURF generates good results in terms of triangulation error, an average increase of 21% in the number of matched features is achieved when ORB is employed. Furthermore, the lowest standard deviation and the least variability in the error range and distribution among patients is reached with ORB features. This observation can lead to a further improvement where, in an ideal system, the best detectors for each frame are automatically selected to achieve the minimal error.

As, SURF, MSER, and ORB achieve the highest number of matched inliers, (Table 2), a combination of SURF-ORB features into one vector with location and metric information of the image feature, may lead to a higher matching accuracy. However, when SURF is employed, the minimum number of matched features among all frames can be as low as only two. Two matched feature points will guarantee to estimate the vertebrae transformation, as described in [30].

This study demonstrates that the presented algorithms offer an accurate and attractive solution to replace currently used DRFs and markers, for navigated spine surgery.

## 5. Limitations

Despite the above advantages, the experiments presented in this study were subject to poor illumination conditions that can affect performance. Those conditions may cause inaccurate matching. It should be noted that the current acquisition does not operate with real-time video, meaning that the algorithm is applied for a pair of frames, for each time stamp. In an in-vivo scenario a live video should be used for spine feature detection during surgery. In future real-time applications, technical aspects such as the illumination conditions in the operation room should be more controlled. Further, the results of this study were based on 23 retrospective clinical cases. Addition of more cases would strengthen the conclusions. A prospective protocol, comparing skin markers to spine feature detection should be considered for future studies. Another limitation when using spine features for spinal navigation is the issue of visibility of anatomy. Obviously, spine features cannot be used in minimally invasive procedures due to the nature of the surgery. However, a considerable number of surgeries are still performed openly and offer a view of the posterior aspects of the spine to be used for detection and tracking. Still, care must be taken to not obscure the camera view by retractors and other surgical instruments. Blood may also obscure spinal features and interfere with detection, especially when gray scale images are used. However, achieving adequate homeostasis and a clean surgical site is arguably part of the routine in navigated spine surgery. Feature detection can possibly also be improved by employing color or hyperspectral cameras.

## 6. Conclusions

In this study, a computer vision framework was created for preprocessing of the spine images, detecting and matching local invariant image regions. Four feature detection algorithms, SURF, MSER, FAST, and ORB were compared to elucidate the best approach. While SURF generates the best results in terms of 3D matching error, ORB achieves the highest number of matched features. Therefore, a combination of SURF-ORB feature descriptors would achieve a higher accuracy. The framework was validated in 23 patients and the 3D triangulation error of the matched features was <0.5 mm. Thus, the findings indicate that spine feature detection can be used for accurate tracking in navigated surgery. For future work, the aim would be to reconstruct and evaluate the intervertebral movement for a more accurate intraoperative navigation. For this purpose, higher resolution images are required to clearly visualize individual vertebrae. This information can be fused with intraoperative CT scans, to ensure the correct correspondences along the spine. By knowing the location of two feature points per vertebra, the similarity transformation can be easily computed, and the spine movements corrected during the surgical procedure. We conclude that the results of this study serve as a robust basis for development of a spine tracking software without using any indirect markers, thereby simplifying the clinical workflow.

## Figures and Tables

**Figure 1 sensors-20-03641-f001:**
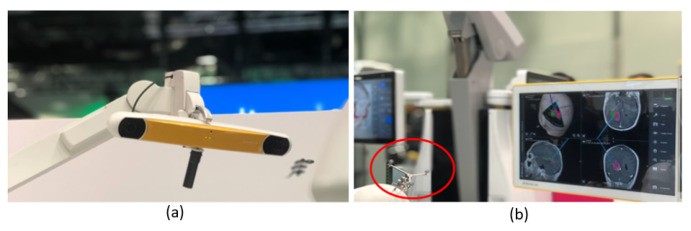
Images of standard setup with indirect patient tracking ( [18]). (**a**) The stereoscopic infrared camera setup for tracking the DRF (dynamic reference frame). (**b**) The DRF (in the red circle) and the monitor displaying the position on a CT patient scan.

**Figure 2 sensors-20-03641-f002:**
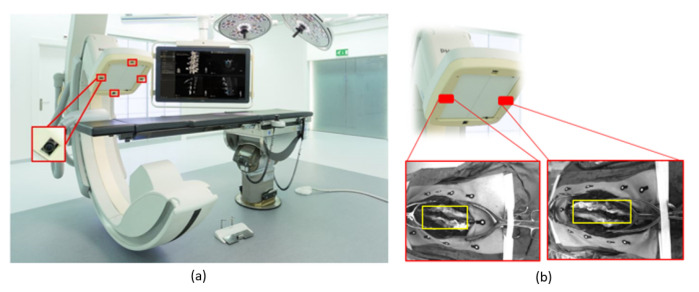
(**a**) The optical tracking system, four optical cameras are integrated in the flat panel of the X-ray detector (Philips Electronics B.V., Best, The Netherlands). (**b**) The optical cameras we used in this preliminary study.

**Figure 3 sensors-20-03641-f003:**
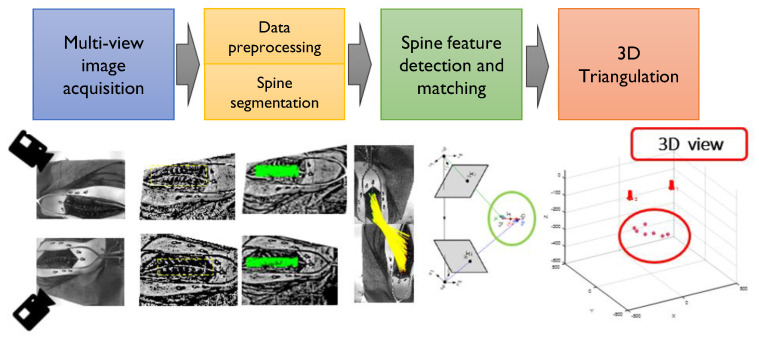
Proposed framework for 3D spine feature localization.

**Figure 4 sensors-20-03641-f004:**
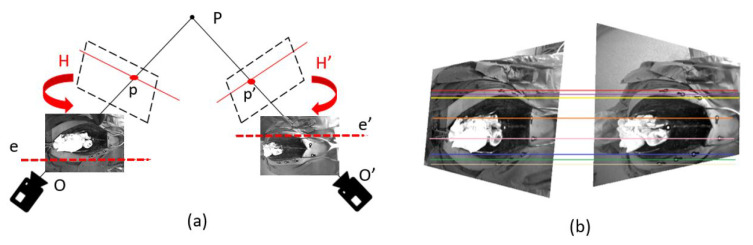
Image rectification, (**a**) the projective transformation *H* lead the matching of the epipolar lines. (**b**) shows an epipolar geometry matching after image rectification, the epipolar lines plotted for the optical markers run parallel after image rectification.

**Figure 5 sensors-20-03641-f005:**
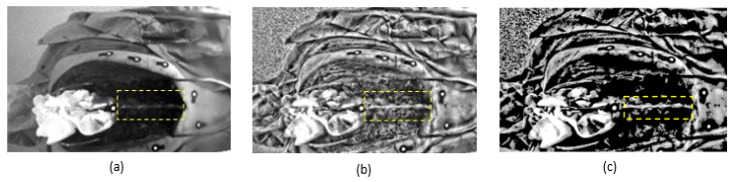
Image preprocessing to improve feature detection. (**a**) Original image view, (**b**) result after preprocessing, (**c**) spine segmentation after image thresholding indicated by the dotted box.

**Figure 6 sensors-20-03641-f006:**
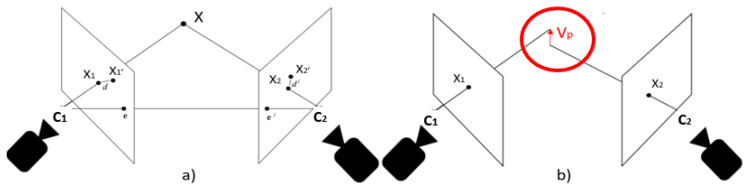
(**a**) An ideal triangulation where $C_{1}$ and $C_{2}$ define the 3D positions of Cameras 1 and 2, respectively. The 3D Triangulation of point *X* is projected into two camera views in the points $x_{1}$ and $x_{2}$, (**b**) $V_{p}$ is the line segment representing the shortest distance between to two back-projected lines.

**Figure 7 sensors-20-03641-f007:**
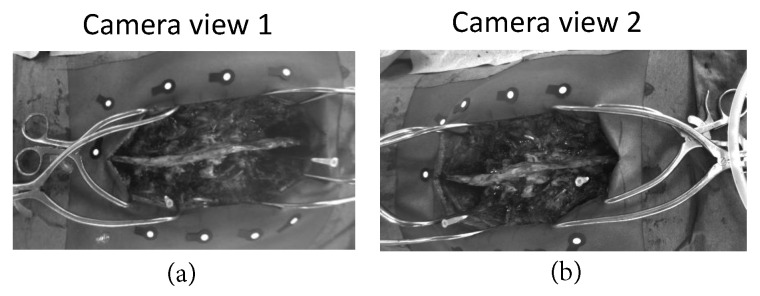
A pair of frames, captured by the camera view 1 (**a**) and the camera view 2 (**b**) for a unit of time, is visualized.

**Figure 8 sensors-20-03641-f008:**
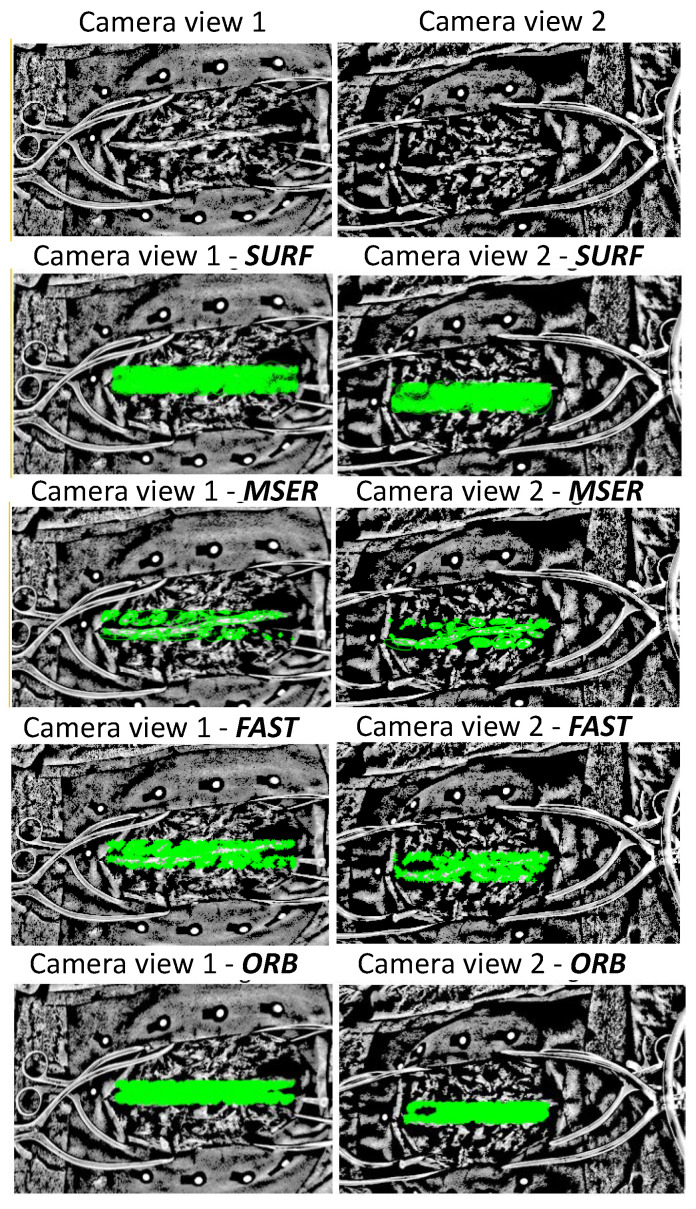
Detection of spine features in two image views (camera 1 and 2). The numbered views above the two columns of images, refer to the two cameras. In the first row the preprocessed images are shown. The subsequent rows illustrate results after applying the four feature detection algorithms (SURF, MSER, FAST and ORB). Green is used to indicate salient spine features identified by the respective algorithms.

**Figure 9 sensors-20-03641-f009:**
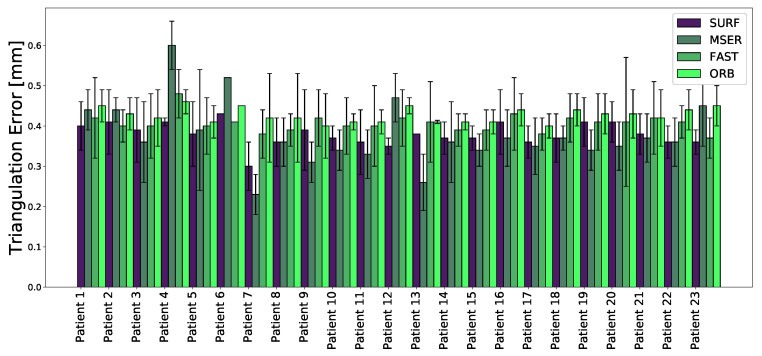
Mean 3D triangulation error and standard deviation for different local feature descriptors calculated per patient.

**Figure 10 sensors-20-03641-f010:**
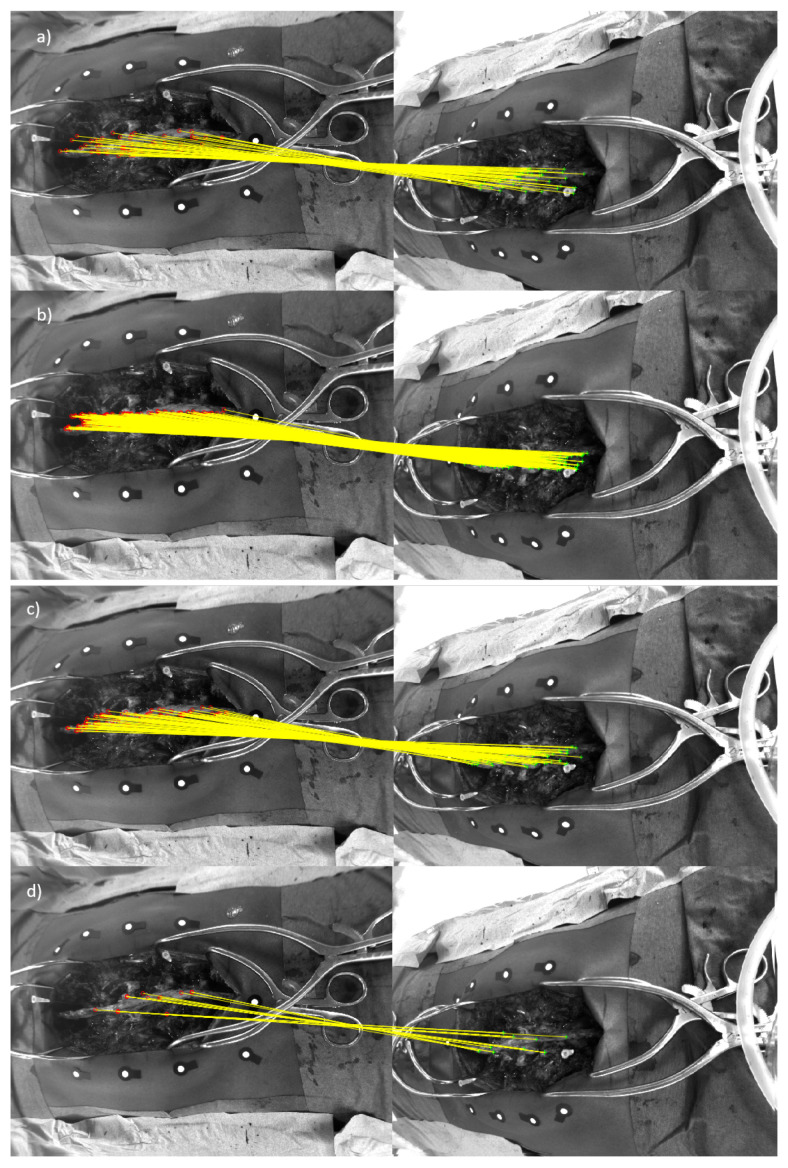
(**a**) FAST-based feature matching of the spine for two image views for Patient 15. (**b**) ORB-based feature matching of the spine for two image views for Patient 15. (**c**) SURF-based feature matching of the spine for two image views for Patient 15. (**d**) MSER-based feature matching of the spine for two image views for Patient 15.

**Figure 11 sensors-20-03641-f011:**
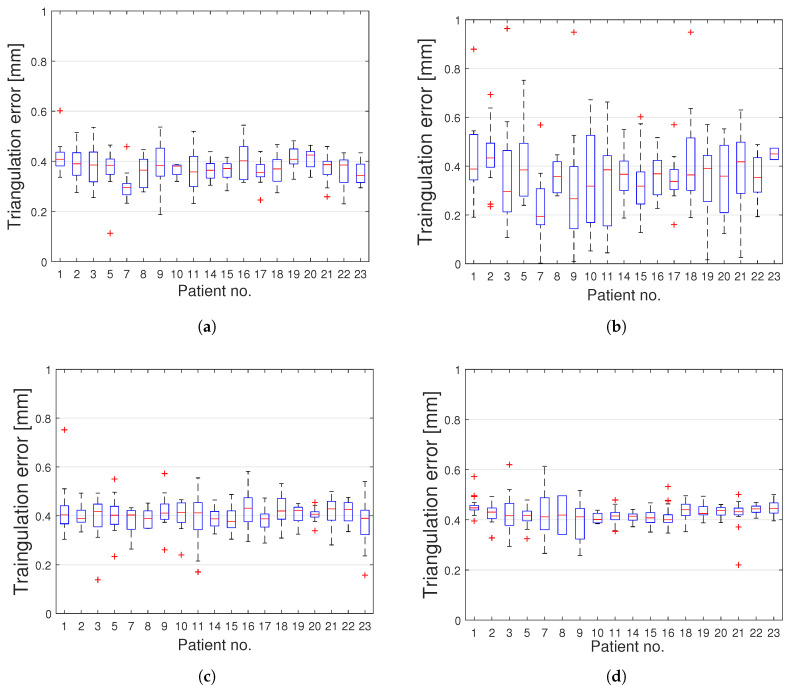
Boxplots comparing the total mean triangulation error in 23 patients when using SURF (**a**), MSER (**b**), FAST (**c**), ORB (**d**) as feature detectors. Upper and lower limits of the box represent 75th and 25th percentiles, respectively. The median is represented by a line transecting the box. Whiskers represent the max and min values. Outliers are plotted using the ‘+’ symbol.

**Figure 12 sensors-20-03641-f012:**
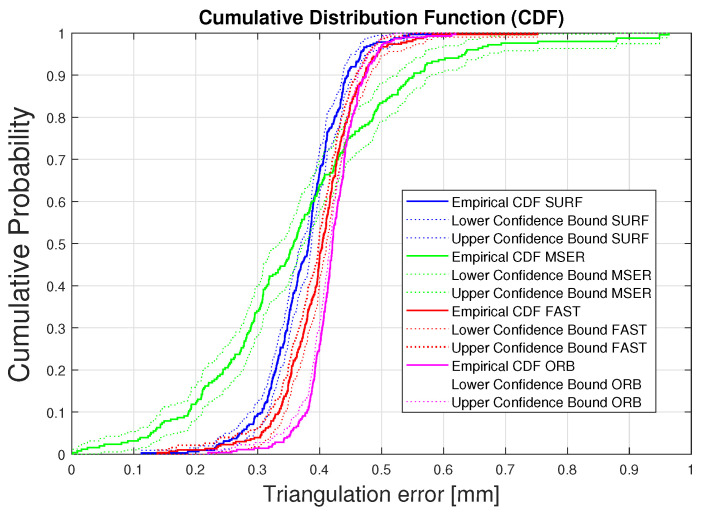
Cumulative distribution function (CDF) of the triangulation error for the four methods. The dotted area delimits the 95% confidence interval for each CDF. The triangulation error of the analyzed frames is lower than 0.5 mm in more than 97% (SURF) or 95% (FAST and ORB). For MSER, however, the corresponding number is 84%, likely explained by the larger variability in triangulation errors.

**Figure 13 sensors-20-03641-f013:**
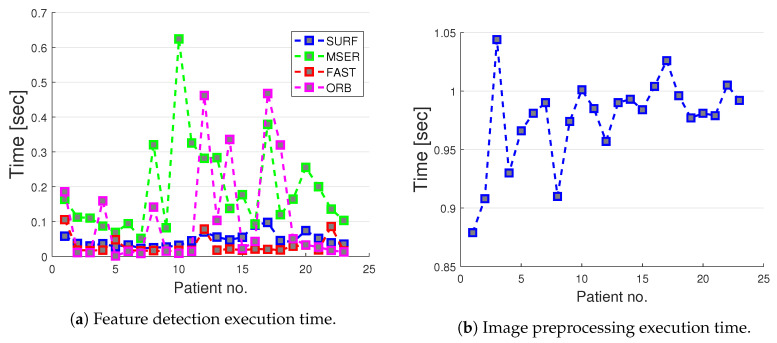
Average execution time for different feature detection methods (**a**) and preprocessing time (**b**) in spine patients.

**Table 1 sensors-20-03641-t001:** Parameters for Speeded Up Robust Feature (SURF), Maximal Stable Extremal Region (MSER), Features from Accelerated Segment Test (FAST), and Oriented FAST and Rotated BRIEF (ORB) feature detectors.

**SURF**	Feature threshold = 600	Number of octaves = 4	Number of scales = 6
**MSER**	Step size for threshold = 0.3	Region size = [100, ..., 800]	Area variation = 0.3
**FAST**	Min. corner quality = 0.1	Min. intensity = 0.2	
**ORB**	Scale factor = 1.2	Decomposition levels = 8	

**Table 2 sensors-20-03641-t002:** Maximum (Max.), Minimum (Min.), Mean, Median, and Interquartile Range (IQR) of matched features on all patients after outlier removal per unit of time (pair of frames).

	Max. Inliers	Min. Inliers	Mean Inliers	Median Inliers	IQR Range
	No.	No.	No.	No.	No.
**SURF matched features**	177	2	40	26	42.25
**MSER matched features**	214	1	15	25	30
**FAST matched features**	94	2	28	33	29
**ORB matched features**	732	1	131	181	209.5

**Table 3 sensors-20-03641-t003:** Mean and standard deviation of execution time for different feature detection methods, and preprocessing on all patients (pair of frames).

Patient No.	Execution Time (Mean ± std)				
	**SURF [sec]**	**MSER [sec]**	**FAST [sec]**	**ORB [sec]**	**PREPROC. [sec]**
**1**	0.058 (±0.024)	0.164 (±0.056)	0.105 (±0.220)	0.185 (±0.460)	0.879 (±0.314)
**2**	0.037 (±0.005)	0.113 (±0.032)	0.018 (±0.002)	0.011 (±0.005)	0.908 (±0.276)
**3**	0.030 (±0.006)	0.110 (±0.032)	0.017 (±0.002)	0.011 (±0.008)	1.044 (±0.185)
**4**	0.036 (±0.013)	0.087 (±0.021)	0.018 (±0.003)	0.159 (±0.452)	0.930 (±0.223)
**5**	0.028 (±0.010)	0.069 (±0.014)	0.048 (±0.044)	0.005 (±0.008)	0.966 (±0.007)
**6**	0.033 (±0.011)	0.094 (±0.041)	0.018 (±0.001)	0.015 (±0.012)	0.981 (±0.020)
**7**	0.023 (±0.006)	0.052 (±0.012)	0.016 (±0.003)	0.008 (±0.003)	0.990 (±0.025)
**8**	0.025 (±0.006)	0.320 (±0.280)	0.016 (±0.300)	0.141 (±0.500)	0.910 (±0.282)
**9**	0.027 (±0.005)	0.082 (±0.120)	0.015 (±0.010)	0.015 (±0.200)	0.974 (±0.030)
**10**	0.032 (±0.004)	0.032 (±0.400)	0.017 (±0.320)	0.009 (±0.427)	1.001 (±0.039)
**11**	0.045 (±0.009)	0.045 (±0.510)	0.018 (±0.300)	0.015 (±0.500)	0.985 (±0.017)
**12**	0.070 (±0.015)	0.070 (±0.510)	0.078 (±0.220)	0.461 (±0.990)	0.957 (±0.021)
**13**	0.055 (±0.025)	0.055 (±0.380)	0.018 (±0.001)	0.103 (±0.380)	0.990 (±0.016)
**14**	0.047 (±0.020)	0.047 (±0.070)	0.021 (±0.007)	0.335 (±0.600)	0.993 (±0.032)
**15**	0.055 (±0.008)	0.177 (±0.007)	0.018 (±0.002)	0.022 (±0.002)	0.984 (±0.001)
**16**	0.090 (±0.004)	0.093 (±0.005)	0.021 (±0.002)	0.043 (±0.001)	1.004 (±0.050)
**17**	0.097 (±0.020)	0.379 (±0.360)	0.020 (±0.020)	0.467 (±0.060)	1.026 (±0.096)
**18**	0.045 (±0.013)	0.120 (±0.060)	0.018 (±0.002)	0.319 (±0.031)	0.996(±0.028)
**19**	0.041 (±0.011)	0.164 (±0.031)	0.029 (±0.038)	0.050 (±0.108)	0.977 (±0.042)
**20**	0.073 (±0.026)	0.255 (±0.077)	0.032 (±0.044)	0.032 (±0.014)	0.981 (±0.032)
**21**	0.051 (±0.020)	0.199 (±0.078)	0.018 (±0.002)	0.026 (±0.014)	0.979 (±0.024)
**22**	0.038 (±0.011)	0.135 (±0.051)	0.085 (±0.269)	0.016 (±0.008)	1.005 (±0.026)
**23**	0.035 (±0.008)	0.103 (±0.001)	0.015 (±0.003)	0.012 (±0.006)	0.992 (±0.011)

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
