# Peer review of "Towards Optical Imaging for Spine Tracking without Markers in Navigated Spine Surgery"

_sensors, 2020, doi:10.3390/s20133641_

Round 1

Reviewer 1 Report

The manuscript studies spine tracking without markers in navigated spine surgery. 

(1) The contributions needs to be clarified.

(2) There are a lot of papers about the tracking without markers in navigated surgery. The authors need to clarify the advantages of the proposed method and compare the proposed method with the existing tracking methods without markers.

Reviewer 2 Report

The paper is interesting, but some aspects could be clarified. I would recommend publication after major revisions.   1. Please re-write the contribution section to point out the main difference between your work and the previous ones? It is not clear to me what is the contributions of your work.   2. The paper discussed the strength of the proposed work (e.g., usage of noninvasive method) but did not clearly compare it to the state-of-the-art.  The authors need to expand the related work section, and experimentally compare their work to the state-of-the-art methods.    3. Section 1: The details and key points of the proposed approach should be pointed out.   4. The comparison with the state-of-the-art should be improved to clarify the novelties also considering the experimental findings.   5. The Conclusions are insufficient for a journal paper. The authors claims should be discussed and highlighted in the conclusion. Also, future works to address the current limitations of the paper should be clearly discussed.    6. I suggest to report the result using several metrics instead of using a specific metric, unless it is justified.    7. The Reference style should be strictly compliant with the MDPI guidelines. For instance, MDPI requires that "Cited journals should be abbreviated according to ISO 4 rules". Please refer to the following link and abbreviate all the journal names accordingly: https://www.mdpi.com/authors/references   8. There are many writing, grammatical, and punctuation errors  (e.g., line 21, 24, 30, 32, etc).  Several rounds of proofreading should be performed by a native speaker to fix all these errors. 

Reviewer 3 Report

Minor comments:

  1. Check the spelling of the document, punctuation, and the excessive use of words.
  2. Update some references with relevant articles published in standard indexed journals for an introduction that allows the reader to understand the problems that exist and the solutions that have been proposed.
  3. Please extend and reformulate the abstract by telling prospective readers what you did and what the important findings of your research were. The abstract is limited to 150–200 words and provides a short description of the perspective and purpose of your paper. It gives key results but increases experimental details.
  4. When cutting images manually, this should affect the dimensions or features extracted, which can occur with fatal errors in a real situation due to a lack of calibration. It does not mention how to solve this problem and, if possible, because stereo camera settings must be preconfigured.
  5. Please be more precise with the contribution of your research. This document shows the implementation of image processing tools and filters that could be a low contribution for publication in a journal. You must justify the contribution and specify it in more detail.
  6. In the results, you could add the reconstruction of the spine performed by the configuration of the two cameras (3D triangulation) and how the triangulation error was obtained, what methods and references you used to find this data.
  7. Please rewrite the conclusion. This should be a summary of the evidence of the results achieved and their importance. In the document, this appears to be one more section of future work.

Major Concern:

I could not see the scientific claims, the author says "In this work, the feature detection algorithms SURF, MSER, FAST and ORB were extensively compared". There are compared algorithms and implementing technology. However formal theory should be described, a pair of equations is not enough. Probably if the novelty of the work is the comparison, that must be mentioned in the title of this manuscript.

Round 2

Reviewer 1 Report

The revised manuscript has addressed the comments.

Reviewer 2 Report

Thank you for addressing the comments. 

This manuscript is a resubmission of an earlier submission. The following is a list of the peer review reports and author responses from that submission.

Round 1

Reviewer 1 Report

This paper introduces a comparison of four different algorihtms to test the feasibility of detecting and matching spinal features using multi-view stereo images.

The paper is well-written and every introduced concept/method is easy to follow. Probably, more details about the four methods included in the comparison would shed some light about their differences in the reported performance. 

Regarding the results, it is true that the mean errors provided by SURF and MSER are the lowest ones, but ORB provides a very low standard deviation (0.02). This makes me think that if you combine (as future work) somehow the different methods, automatically selecting the best model for each case, the accuracy would be even better. 

To have a better view on the individual results provided by each of the four methods on the evaluation cases. Could you please include one plot with the CDF's of the triangulation errors with the four methods?

Reviewer 2 Report

The paper presented a technology to omit the usage of markers. It is an interesting topic and valuable for clinical applications. But, there are some issues need to be addressed:

  1. It is known that the visibility of spine surgery is very limited. Even in the open spinal surgery, the quality of the images is often affected by blood stains and obstacles. It is a major problem which limits the application of surgical navigation technologies with grey images. This problem seems much more important than feature detection. Therefore, it is suggested to supplement some introductions about this problem.
  2. Another problem is about the real-time performance, which is related to feasibility of practical application. The preprocessing and feature detecting of grey images take some time, which results in the decrease in real-time performance. This is one of the reasons why the mark-point-based navigation technologies are more popular currently. What contributions does this paper make for improving the real-time performance?
  3. 18 open spinal surgery cases were employed to verify the accuracy of the technology. More cases are needed. In addition, the gold standard should adopt the measurements of internationally recognized navigation equipment, or professional measuring equipment.

Round 2

Reviewer 2 Report

The revised manuscript is better than the last version. However, the improvement is not enough.

1) The novelty is limited.

2) The experiments are not enough.